# Ready for handwriting? A reference data study on handwriting readiness assessments

Helga Haberfehlner[1], Liesbeth de Vries[2,3], Edith H. C. Cup[4], Imelda J. M. de Groot[4], Maria W. G. Nijhuis-van der Sanden[5], Margo J. van Hartingsveldt[2]*

1 Department of Rehabilitation Sciences, KU Leuven, Campus Bruges, Bruges, Belgium, 2 Research Group Occupational Therapy, Urban Vitality, Centre of Expertise, Faculty of Health, Amsterdam University of Applied Sciences, Amsterdam, The Netherlands, 3 Occupational Therapy Programme, Hanze University of Applied Sciences, Groningen, The Netherlands, 4 Department of Rehabilitation, Donders Institute for Brain, Cognition and Behaviour, Radboud University Medical Center, Nijmegen, The Netherlands, 5 Research Institute for Health Sciences, IQ healthcare, Radboud University Medical Center, Nijmegen, The Netherlands

* m.j.van.hartingsveldt@hva.nl

**Data Availability Statement:** https://doi.org/10.21943/auas.19236195.v1.

**Funding:** The author(s) received no specific funding for this work.

## Abstract

### Introduction

Early evaluation of writing readiness is essential to predict and prevent handwriting difficulties and its negative influences on school occupations. An occupation-based measurement for kindergarten children has been previously developed: Writing Readiness Inventory Tool In Context (WRITIC). In addition, to assess fine motor coordination two tests are frequently used in children with handwriting difficulties: the modified Timed Test of In-Hand Manipulation (Timed TIHM) and the Nine-Hole Peg Test (9-HPT). However, no Dutch reference data are available.

### Aim

To provide reference data for (1) WRITIC, (2) Timed-TIHM and (3) 9-HPT for handwriting readiness assessment in kindergarten children.

### Methods

Three hundred and seventy-four children from Dutch kindergartens in the age of 5 to 6.5 years (5.6±0.4 years, 190 boys/184 girls) participated in the study. Children were recruited at Dutch kindergartens. Full classes of the last year were tested, children were excluded if there was a medical diagnosis such as a visual, auditory, motor or intellectual impairment that hinder handwriting performance. Descriptive statistics and percentiles scores were calculated. The score of the WRITIC (possible score 0–48 points) and the performance time on the Timed-TIHM and 9-HPT are classified as percentile scores lower than the 15th percentile to distinguish low performance from adequate performance. The percentile scores can be used to identify children that are possibly at risk developing handwriting difficulties in first grade.

**Competing interests:** The authors have declared that no competing interests exist.

## Results

WRITIC scores ranged from 23 to 48 (41±4.4), Timed-TIHM ranged from 17.9 to 64.5 seconds (31.4± 7.4 seconds) and 9-HPT ranged from 18.2 to 48.3 seconds (28.4± 5.4). A WRITIC score between 0–36, a performance time of more than 39.6 seconds on the Timed-TIHM and more than 33.8 seconds on the 9-HPT were classified as low performance.

## Conclusion

The reference data of the WRITIC allow to assess which children are possibly at risk developing handwriting difficulties.

## Introduction

Despite the increasing use of computers, tablets, and smartphones, handwriting remains an important skill that children learn to participate in school [1]. The brain has close functional relationships between the reading and writing processes [2]. James and Engelhart stated that handwriting is important for the early conscription in letter processing of brain regions known to underlie successful reading and may facilitate reading achievement in young children [3]. Although the time children spent on fine motor activities and handwriting in school has decreased over the last 20 years as technology has become more important in childhood education, primary school children still spend 18%-47% of classroom activities on fine motor activities mainly handwriting [4]. Therefore, prewriting and handwriting remain an important goal of education in primary school. In the Netherlands children learn prewriting skills in kindergarten at an age of 5 to 6 years. In this phase, children learn an appropriate sitting posture, to handle the pencil properly and produce different writing patterns before they start handwriting with cursive or block letters in first grade. In the Netherlands, children enter first grade normally in the school year in which they turn six before 1$^{st}$ of January. However, teachers can decide together with parents to let children duplicated the last kindergarten year due to developmental or social delays. Learning to write is not easy for everyone, some children develop handwriting difficulties. The prevalence of handwriting difficulties in 6–12 old children ranges between 6% and 33% [5]. It has been reported that children with handwriting difficulties develop negative experiences in this area, including frustration, self-reliance, and low motivation [6]. Persistent handwriting difficulties may also have negative effects on a child's academic performance and self-esteem [7]. Identification of kindergarten children at risk of developing handwriting difficulties may allow early intervention to prevent handwriting difficulties and negative secondary effects in later grades.

The Writing Readiness Inventory Tool In Context (WRITIC) is developed to evaluate handwriting readiness in 5–6.5 years old children [8]. In previous group studies, it was found that the WRITIC is a valid, reliable, feasible and predictive measure [8–10]. The WRITIC gives valuable criterion-referenced information on handwriting readiness by evaluating factors regarding the child, the environment and the paper-and-pencil tasks [8]. The subdomain 'Task performance' of the WRITIC (WRITIC-TP) will be the norm-referenced subdomain which clarifies if children are ready to learn the skill of handwriting [8]. However, reference data are not yet available.

If children are not ready for handwriting, it has been suggested assessing performance components of handwriting. This assessment should include fine motor coordination [9, 11, 12] and visual-motor integration [6, 9, 13, 14] since these components have been found to be

prerequisites for handwriting [13]. Visual-motor integration can be tested by the Beery-Bukte-nica Developmental Test of Visual-Motor Integration (Beery VMI) [15]. For measuring fine motor coordination two tests have been found most feasible The first is the Nine-Hole Peg Test (9-HPT) [16, 17], which evaluates static fine motor skills is widely accessible [18]. The second is the Test of In-Hand Manipulation Revised (TIHM-R) [19]. The TIHM-R, was recently modified in a Dutch sample to the Timed Test of In-Hand Manipulation (Timed TIHM) [20], which evaluates the dynamic fine motor skills of in-hand manipulation. For the Beery VMI applicable reference values are available [15], while for the 9-HPT only American reference values exist for the age group of 5–6.5 years [16, 17]. For the recently modified Timed-TIHM no reference values are available.

The aim of the current study was to provide reference data for (1) WRITIC-TP, (2) Timed-TIHM and (3) 9-HPT for the assessment of handwriting readiness in kindergarten children (age 5–6.5 years). In addition, gender and age differences on WRITIC-TP, Timed-TIHM and 9-HPT were assessed and the correlations between these three tests were evaluated. Based on earlier research [8, 10, 20] our hypothesis is that both fine motor tests would have a moderate correlation with WRITIC-TP.

## Materials and methods

### Participants

Children in the age of 5 to 6.5 years were recruited by asking 25 teachers of kindergarten classes of elementary schools for their participation. In the Netherlands children perform prewriting skills in kindergarten, before they start in grade one with un-joined cursive script or block letters. Full classes were tested. Schools from different parts of the Netherlands were approached in order to represent the present cultural diversity of children in the Netherlands. Migration background was collected to control for a representative distribution within our sample. Following personal characteristics of the child were registered: age rounded to full months, gender and dominate hand.

Ethical approval has been obtained from the Institutional Review Board of the Research Committee involving Human Subjects (Dutch abbreviation: CMO) of Radboudumc (Arnhem-Nijmegen region) under IRB no. 35498.091.11. Prior to participation, the parents of each child were informed of the study and asked to sign informed consent. Children were excluded if they were unable to complete the items of the WRITIC due to a medical diagnosis such as a visual, auditory, motor or intellectual impairment.

### Instruments

**Writing Readiness Inventory Tool In Context (WRITIC).** The WRITIC is an occupation-based assessment to evaluate handwriting readiness [8]. The WRITIC contains items of three domains: 'Child', 'Environment', and 'Paper-and-pencil tasks'. Every domain consists of two subdomains: the 'Child' domain includes 'Interest' and 'Sustained attention', the 'Environment' domain includes 'Physical environment' and 'Social environment', and the Paper-and-pencil tasks domain includes 'Task performance' and 'Intensity of performance'. The WRITIC is administered in the classroom, where the influence of the context is taken into account. First, the child's interest in paper-and-pencil tasks is evaluated. Then, the child is encouraged to complete a drawing booklet with seven paper-and-pencil tasks while a trained assessor observes and scores performance and quality of the paper-and-pencil tasks.

The subdomain 'Task performance' (i.e., WRITIC-TP) within the 'Paper-and-pencil tasks' domain was developed as a norm-referenced part of the WRITIC. The WRITIC-TP consists of seven items scoring the quality of paper-and-pencil tasks (tracing double-lined paths,

colouring, making arcades, making garlands, name writing, making spirals, and copying letters and numbers) on a 3-point scale (range 0–2, maximum score: 14), and the performance of these tasks regarding sitting posture and pencil grip: five items (type of pencil grip, sitting posture, forearm position, distal versus proximal movement, and other hand) are scored on a 7-point scale with a range of 0 to 6 resulting in a maximum score of 30 and one item (wrist position) on a 5-point scale (range 0–4, maximum score: 4). So, the total score ranges from 0–48.

Previous research of the WRITIC-TP confirmed high internal consistency (Cronbach's alpha = 0.82), significant ability to discriminate between children with good and poor performance on paper-and-pencil tasks (U = 11.78, p < 0.001), and excellent test-retest and inter-rater reliability, with ICC's of 0.92 and 0.95 respectively [8–10]. The WRITIC-TP, administered in kindergarten, is found to be the main predictor for handwriting quality evaluated by the Systematic Screening for Handwriting Difficulties [21] in grade 1 [9]. The WRITIC takes approximately 20 minutes to administer.

**Timed Test of In-Hand Manipulation (Timed TIHM).** The Timed-TIHM assesses three skills of in hand manipulation: 1) translation from finger to palm; 2) translation from palm to finger; and 3) complex rotation of 360˚ [20]. The test is designed for children from 5 to 6 years of age. It takes 5 to 7 minutes to administer. For the Timed-TIHM the commercially available 9-Hole Peg Test Kit is used. The child is asked to pick up successively two, three, four, and five pegs with the dominant hand, manipulate the pegs with the fingertips to the palm, and keep them in the palm of the hand (translation from finger to palm with stabilization), and then to replace the pegs one by one into the pegboard (translation from palm to finger with stabilization). The tasks with two, three, four and five pegs are included as practice items, followed by two trials with five pegs to score the time. The third task is two trials with a complex rotation task in which the child is asked to rotate one peg 360˚ for a total of five times using the fingertips of the dominant hand. The best time score of the two trials is used as the outcome measure, with a low time score corresponding to good fine motor performance. The number of drops and the times on external surface that is used to compensate are recorded as supplemental qualitative information. The time of the two translation and two rotation tasks are added and used for further analysis.

Research with the previous version of this assessment confirmed inter-rater reliability and construct validity using Rasch modeling: the TIHM-Revised [19]. In a Dutch population Test-retest reliability of the current Timed-TIHM was good with an ICC of 0.71. Convergent validity with the WRITIC-TP was moderate with $r_s$ = 0.40 [20].

**Nine-Hole Peg Test (9-HPT).** The 9-HPT evaluates simple fine motor patterns, including reaching, grasping, carrying, entering, and releasing with the time taken to perform these tasks as the outcome measure [16]. The 9-Hole Peg Test Kit is a simple, commercially available timed test of fine motor coordination in which nine pegs are inserted one by one into a pegboard and then consecutively removed. In our research children were asked to complete the task twice with the dominant hand, which they used for paper-and-pencil tasks. The best time score was used, with a low score (less time needed to perform the task) corresponding to good fine motor performance.

The 9-HPT has been validated in an American study population of 826 children between 5 and 10 years of age. High inter-rater and test–retest reliability was established, and strong construct validity was obtained. Normative values are available for children in this age category [16].

## Procedure

Children were assessed on the WRITIC, the Timed-TIHM and the 9-HPT. The WRITIC was administered individually in the classroom during a time when all the children were doing

different tasks in small groups. The Timed-TIHM and the 9-HPT were administered in the same session outside the classroom in a one-to-one situation, in a random order due to possible fatigue. According to the informed consent, children could stop if they indicated.

Test administrators included 27 bachelor students of occupational therapy, exercise therapy and physical therapy. To become competent in administering the tests, all administrators 1) attended a full day training including theory of handwriting readiness and fine motor coordination, practicing scoring the WRITIC, the Timed-TIHM and the 9-HPT by means of videos of assessments and practicing administration of the three tests on each other; 2) practised WRITIC with two typically developing children; 3) assessed the first two children in the presence of another student in order to give each other feedback and guarantee consistence between the test administrators.

## Statistical analysis

Raw data of WRITC-TP, Timed-TIHM and the 9-HPT were used for statistical analysis. Normal distribution was checked by means of a normal probability plot and tested by Kolmogorov-Smirnov test of normality. Children were divided in age groups of each 6 months (i.e.: 60–65 months = 5-.5.5 years; 66–71 months = 5.5–6 years; 72–78 months = 6–6.5 years). Scores of all three tests were compared among age groups and gender. Descriptive statistics and percentiles scores were calculated. Non-parametric tests (i.e., Mann-Whitney U Test to test for gender differences and Kruskal-Wallis Test to test for differences in age groups) were used, as data was not normally distributed.

Correlations between the three tests were calculated by Spearman's *rho* ($\rho$). 001). The criteria proposed by Hinkle and colleagues were used for interpretation: a correlation coefficient greater (lower) than 0.90 (-0.90) was considered very high positive (negative) correlation, 0.70 to 0.90 (-0.70 to -0.90) high positive (negative) correlation, 0.50 to 0.70 (-0.50 to -0.70) moderate positive (negative) correlation, 0.30 to 0.50 (-0.30 to -0.50) low positive (negative) correlation and 0.00 to 0.30 (-0.00 to -0.30) little if any correlation [22].

The group of children that scored on one of the three test lower than the 15th percentile were separately analysed using a VENN-diagram to assess the overlap in scores on WRITIC-TP, 9-HPT and Timed-TIHM [23]. Statistical analyses were performed using SPSS 22 (IBM SPSS Statistics, Amsterdam, The Netherlands). A p-value of less than.05 was considered to indicate a statistically significant difference.

## Results

Three hundred and seventy-four children with a mean age of 5.6 years (SD 0.4) (190 boys, 184 girls) participated in the study. Three hundred and nineteen children were assessed on all three tests, while for 55 only the WRITIC was administered due to time constrains. 87% of the participants was right-handed, 12% left-handed and 1% variable-handed. In the younger age groups (5–5.5 years and 5.5–6 years) there were more children compared to the oldest group (6–6.5 years) (i.e., 42%, 44% and 14%, respectively). Demographics for the age groups are shown in Table 1.

The mean WRITIC-TP was 41.3 points (95% CI = 40.8–41.7), not normally distributed (p<0.001, skewness: -1.05, kurtosis: 1.23) with an SD of 4.4 points. Also 9-HPT (mean±SD: 28.4±5.4 seconds, 95% CI = 27.8–29.0) and Timed-TIHM (mean±SD: 31.4±7.4 seconds, 95% CI = 30.4–32.1) were not normally distributed but positively skewed (p<0.001, skewness of 0.91, kurtosis: 0.76 and p<0.001, skewness = 1.16, kurtosis = 1.86, respectively) (Fig 1).

As data were not normally distributed, non-parametric statistics were applied. Note that for the Timed-TIHM the time value of all children that could perform the test is used. 4.1% of

**Table 1. Descriptive data of participants tested by the WRITIC only (n = 55) and by all three tests (WRITIC, 9-HPT and Timed-TIHM (n = 319).** Temperature and wildlife count in the three areas covered by the study.

| | | Age category (Number (%)) | | | Total group |
|---|---|---|---|---|---|
| | | 5–5.5 years | 5.5–6 years | 6–6.5 years | |
| **Number of participants** | **WRITIC only** | 156 | 164 | 54 | 374 |
| | *All three tests** | *154* | *138* | *27* | *319** |
| **Boys / girls** | **WRITIC only** | 82 / 74 | 83 / 81 | 25 / 29 | 190 / 184 |
| | | (52.6% / 47.4%) | (50.6% / 49.4%) | (46.3% /53.7%) | (50.8% / 49.2%) |
| | *All three tests** | *81 / 73* | *69 / 69* | *14 / 13* | *164 / 155* |
| | | *(52.6% / 47.4%)* | *(50.0% /50.0%)* | *(51.4% /48.1%)* | *(51.4% / 48.6%)* |
| **Right / Left / variable-handed** | **WRITIC only** | 134 /22/0 | 145 / 18 / 1 | 43 / 4 / 2 | 327/ 44 / 3 |
| | | 85.9% / 14.1%/ 0%) | (88.4% / 11.0% / 0.6%) | (87.8% / 8.2% / 4.1%) | (87.4% / 11.8% / 0.8%) |
| | *All three tests** | *133 / 21 / 0* | *120 / 17 / 1* | *24 / 2 / 1* | *277 / 40 / 2* |
| | | *(86.4% / 13.6% / 0%)* | *(87.0% / 12.3% / 0.7%)* | *(88.9% / 7.4% / 3.7%)* | *(86.8% /12.5%/0.6%)* |
| **Non-/ Migration background** | **WRITIC only** | 146 / 10 | 139 /25 | 31 / 23 | 316 / 58 |
| | | (93.6% / 6.4%) | (84.8% / 15.2%) | (57.4% / 42.6%) | (84.5% /15.5%) |
| | *All three tests** | *146 / 8* | *130 /8* | *25 / 2* | *301 / 18* |
| | | *(94.8% / 5.2%)* | *(94.2% / 5.8%)* | *(92.6% / 7.4%)* | *(94.4% / 5.6%)* |

WRITIC = Writing Readiness Inventory Tool In Context, 9-HTP = Nine-Hole Peg Test, Timed-TIHM = Timed Test of In-Hand Manipulation, % = percentage; *Note that only 319 children were assessed on all three tests (cursive = subpopulation of total group, n = 374.

children (n = 13 out of 319 children) could not perform the rotation task of the Timed-TIHM and had as a result no time recorded on this test. The percentile scores of the Timed-TIHM were corrected for these 4.1% of children who were not able to perform the Timed-TIHM completely i.e., they were counted within the group with the lowest scores. The score of the WRITIC-TP and the performance time on the 9-HPT and Timed-TIHM that are classified as percentile scores lower than the 5th percentile, lower than the 15th percentile respectively, as well as the points and time that are classified higher than 15th percentile are displayed in Table 2.

Boys and girls performed slightly different on the WRITIC-TP with girls achieving more points on average on the WRITIC-TP (girls: 42.1±3.8 points; boys: 40.5±4.8 points; p = 0.001).

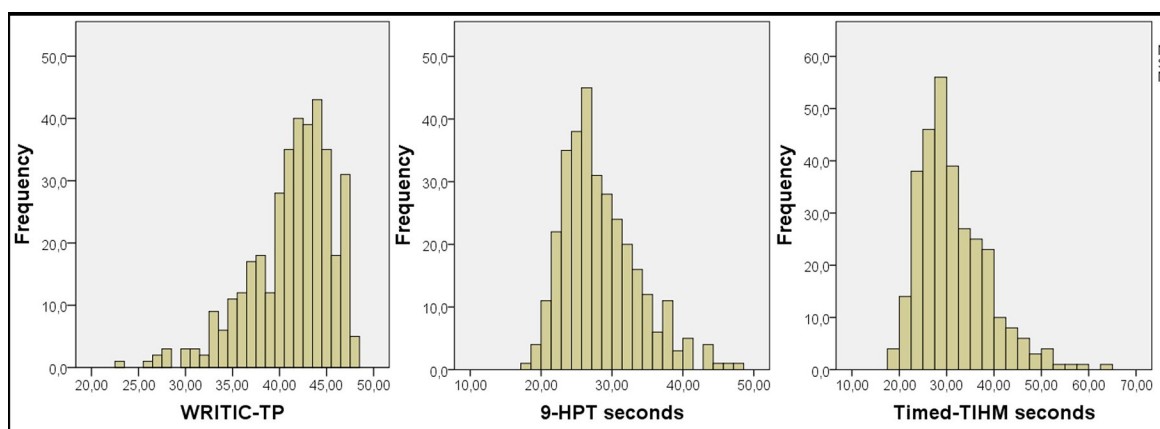

**Fig 1.** Frequency of points on (A) 'Task performance' of the Writing Readiness Inventory Tool In Context (WRITIC-TP), (B) seconds on Nine-Hole Peg Test (9-HPT) and (C) the Timed Test of In-Hand Manipulation (Timed-TIHM). WRITIC-TP is negatively skewed, 9-HPT and Timed-TIHM positively.

**Table 2. Percentile scores of the subdomain 'task performance' of the Writing Readiness Inventory Tool In Context (WRITIC-TP), the Nine-Hole Peg Test (9-HTP) and the Timed Test of In-Hand Manipulation (Timed-TIHM).**

|  | WRITIC-TP | 9-HPT | Timed-TIHM |
|---|---|---|---|
| Percentile score | Raw score | Time | Time |
| <5th percentile | 0–32 points | >38.3 seconds | >56.0 seconds or not able to perform |
| 5th-15th percentile | 33–36 points | 38.3–33.8 seconds | 56.0–39.6-seconds |
| 15th-50th percentile | 37–41 points | 33.9–27.3 seconds | 39.7–30.1 seconds |
| 50th– 85th percentile | 42–44 points | 27.4–23.2 seconds | 30.2–24.5 seconds |
| 85th -95th percentile | 45–46 points | 23.3–21.4 seconds | 24.6–22.2 seconds |
| >95th percentile | 47–48 points | <21.4 seconds | <22.2 seconds |

This difference between boys and girls on the WRITIC-TP was mainly determined by the difference in sum score of the items scoring the quality of results of the paper-and-pencil tasks (p<0.001), but not in the items scoring the quality of performance of these tasks (e.g., sitting position, pencil grip) (p = 0.732). On the Timed-TIHM also a difference was found between boys and girls, with girls performing on average faster (girls: 30.3±6.9 seconds; boys: 32.4±7.7 seconds; p = 0.006). No difference was found on the 9 HPT (p = 0.281). For age groups we found no difference for the WRITIC-TP (5-.5.5 years: 40.9±4.4 points, 5.5–6 years: 41.6±4.6 points and 6–6.5 years: 41.6±3.9 points; p = 0.239), but significant differences on the 9-HPT (5–5.5 years: 29.3±5.7 seconds, 5.5–6 years: 27.7±5.0 seconds and 6–6.5 years: 26.8±4.8 seconds) and Timed-TIHM (5–5.5 years: 32.6±8.1 seconds, 5.5–6 years: 30.1±6.5 seconds and 6–6.5 years: 30.4±6.4 seconds) (p = 0.010 and p = 0.032, respectively), with older children performing faster.

WRITIC-TP, 9-HPT and Timed-TIHM were all correlated: WRITIC-TP had a low negative correlation with the 9-HPT and Timed-TIHM ($\rho$ = -0.285, p<0.001; $\rho$ = -0.331, p<0.001, respectively). 9-HPT had a low positive correlation with the Timed-TIHM ($\rho$ = 0.482, p<0.001).

The group of children that scored on one of the three tests lower than the 15th percentile consisted of 103 children (60 boys). A VENN-diagram in Fig 2 shows the overlap between these subgroups.

Out of the 46 children (30 boys) that scored lower than the 15th percentile on the WRITIC-TP, 21 children (16 boys) out of 46 (45.7%), also had a time score on either the Timed-TIHM or the 9-HPT lower than the 15th percentile, while 8 children (4 boys) out of the 46 (17.4%) had a time lower than the 15th percentile on both other tests.

The full dataset of the current study is available online [24].

## Discussion

With the current research, reference data for the WRITIC-TP, Timed-TIHM and 9-HPT for handwriting readiness assessment in kindergarten children (5 to 6.5 years old) are provided. Gender differences were found for the WRITIC-TP and the Timed-TIHM with girls performing better than boys, but not for the 9-HPT. No difference was found for age groups for the performance on the WRITIC-TP, but it was present for the fine motor coordination assessments (Timed-TIHM and 9-HPT). About half of the children that showed a low performance on the WRITIC-TP, also scored low on fine motor coordination assessed by Timed-TIHM and/or 9-HPT.

The moderate correlation between 9-HPT and Timed-TIHM ($\rho$ = 0.482) in the current study is comparable to previous research [20]. However, the correlation between the WRITIC-TP and the tests of fine motor performance (9-HPT and Timed-TIHM) was lower than

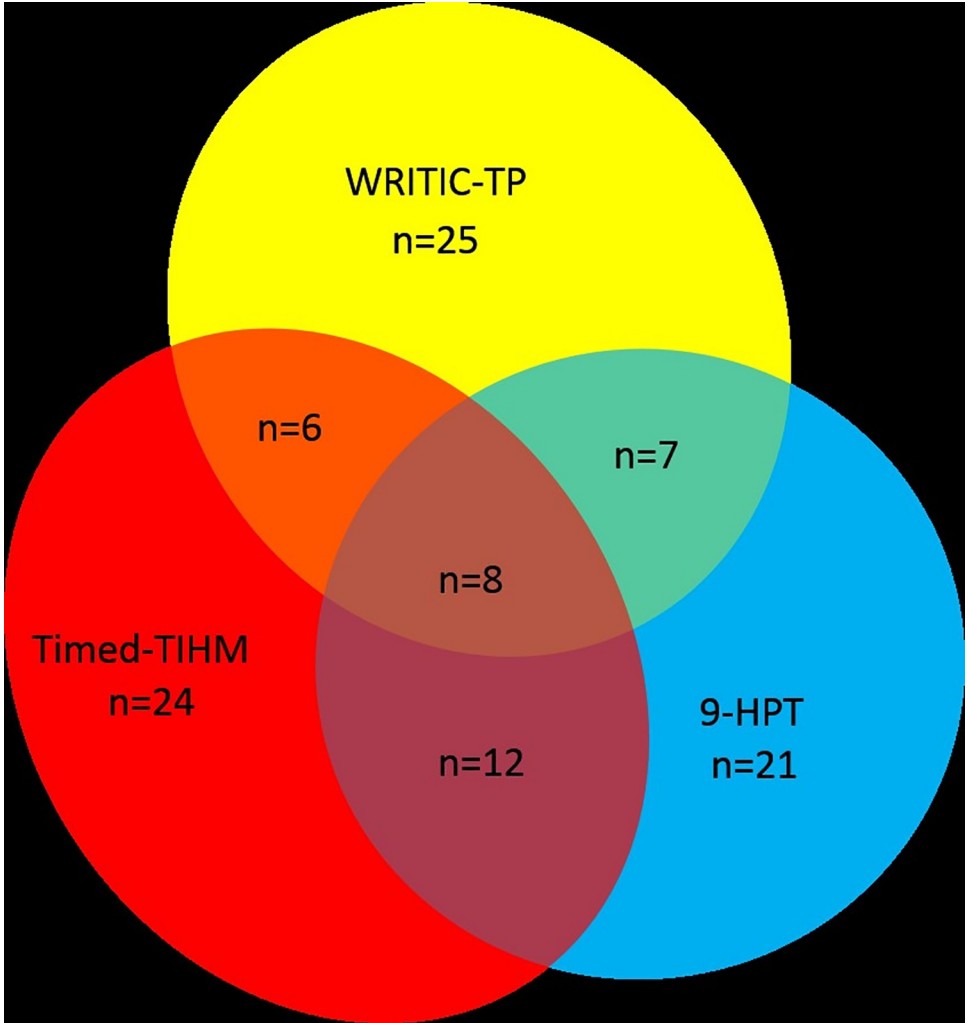

**Fig 2. Group of children (n = 103) that scored lower than the 15th percentile on the subdomain 'task performance' of the Writing Readiness Inventory Tool In Context (WRITIC-TP), on the Nine-Hole Peg Test 9-HPT or the Timed Test of In-Hand Manipulation (Timed-TIHM).** VENN-diagram shows the overlap of children between the three tests.

previously reported, possibly due to a different composition of the population as full classes were tested in the current study. In previous research the participants were selected by the teacher (two or three children with a good and two or three with a poor performance of paper-and pencil tasks according to the teacher) [20].

As expected, not a full overlap in children performing low on all tests as show in the VENN-diagram exists. This finding supports the theory that besides fine motor performance there are other performance components needed for handwriting readiness. In previous research it is reported that besides fine motor coordination also visual-motor integration, as measured with the Developmental Test for Visual-Motor Integration (Beery-VMI) [15], is an important performance component [13, 14, 25]. Besides fine motor coordination and visual-motor integration, sustained attention was also described a predictive factor of handwriting attainment [9, 26, 27].

Although a gender differences on the WRITIC-TP and Timed-TIHM was present in the data, we decided not to provide separate reference values. This decision was based on the fact

that children have to learn handwriting at a certain age, independent of their gender. Interestingly a gender differences for the items of the WRITIC-TP scoring the quality of paper- and pencil tasks, which need practice to achieve proficiency, was found. In contrast, there were no gender differences for the items scoring the quality of performance (e.g., sitting position, pencil grip). Nor did we find differences in gender on the 9-HPT, while the gender differences were evident in the Timed-TIHM. The Timed-TIHM needs more complex fine motor skill that needs practice, in contrast to the basic fine motor skill of the 9-HPT. This is consisted with research into an intervention program on basic foundation skills for handwriting in kindergarten and first grade children. The results indicated a gender effect, with female children improving their fine motor skills more over time than male peers [28]. This finding is line with the research on gender differences in motor learning of Dorfberger and colleagues (2009). The results of their study suggest that gender may be an important factor in motor performance and especially in motor learning. In this study the initial gap in handwriting speed in 9-year old children, in favour of the girls, may represent an advantage in prior experience; however the boys, especially the older participants, are able to close this gap, or even reverse it, when further practice takes place [29].

The reference values of the 9-HPT in the Dutch population are rather comparable with research in an American and Saudi population and shows a decrease in performance time with age [16, 17, 30]. We did not find differences in gender within the age range 5–6.5 year on the 9-HPT, while others found gender differences calculated within a wider age range (i.e., 4–18 years) and with a larger age groups (i.e. one year instead of a half year as used in the current study) [16, 17, 30].

Our reference data are collected in Dutch kindergarten classes and are for this reason limited to this population. In countries with a similar educational system, reference data might be applicable. However, we recommend the collection of reference data within each country separately, as education and daily routines may play a role and show cultural differences. Concerning assessed differences in age groups (i.e., we found that older children were performing faster on the 9-HPT and Timed-TIHM), the results are in the line what is expected. However, no age effect was found for the WRITIC-TP. The results especially for the oldest group (i.e., 6–6.5 years old) should be interpreted with caution taking into account that fewer children were included in this group than in the younger age groups. The reason that fewer children are included in this group was that measurements were performed about six to nine months before children enter grade one (i.e., start learning to write). In addition, some of the older children might have duplicated the last kindergarten year due to developmental delays and therefore show lower performance on the WRITIC-TP. The period (about six to nine months before children enter grade one) to assess handwriting readiness is based on pragmatical reasons, because then there is still enough time for an intervention before the child actually starts learning handwriting [31]. All reference data are presented for the whole age group (5–6.5); the unequal distribution within the age-groups is not relevant in using the reference data in assessing handwriting readiness.

The reference data of the WRITIC-TP show which children are possibly at risk developing handwriting difficulties [9]. If a child is not being ready for handwriting according to the WRITIC-TP and at risk to develop handwriting difficulties in a later grade, additional assessments of performance components (fine motor coordination and visual-motor integration) can be used as second step in the evaluation of handwriting readiness. Based on this two-step assessment interventions to support the child's participation can be planned. Adaptations in the child's physical and social environment such as a pencil grip, a different place in the classroom or an adapted instruction, task-oriented training with enough practicing time and

demonstrating strategies that enhance participation of children in paper-and-pencil tasks have been shown to be effective [32, 33].

A potential limitation of this study is the use of 27 different raters: there could be bias between them. However, the WRITIC-TP has shown an excellent inter-rater reliability within a group of students trained in the same way than in the current study (ICC:0.95;$p < 0.001$) [10]. Moreover, using different raters in this study fits the situation in practice. The students who collected the data attended a full day training on administering WRITIC, practised WRITIC with two typically developing children, and checked their first two administrations with a colleague. To become a reliable administrator, it is essential to follow this training.

## Conclusions

The reference data of the WRITIC-TP show which children are possibly at risk developing handwriting difficulties. The reference data of the 9-HPT and Timed-THIM, as fine motor coordination tests, can be used as a second step in the evaluation of handwriting readiness. We recommend using this kindergarten assessment to assist in achieving the goal of timely intervention for 5- and 6-year-old children and thus prevent handwriting difficulties in later grades.

## Acknowledgments

The authors wish to thank the students who administered the tests in the children: Laudia Borsch, Jill de Haan, Anna Hildebrand, Kelly Huigsloot, Danielle Meijer, Lianne Stark, Michelle van Damme, Carina Dubbeldam, Naära Tomasowa (former students of the bachelor 'Occupational therapy' of Amsterdam University of Applied Sciences) and Aline Averesch, Iris Steijn, Jessica Spaan, Lisan Brookhuis, Lisette Weijers, Michiel van Lingen, Myrthe Verhoog, Mitchell Walgien, Ronalys de Gier, Siem Matton and Tom Smits (former students of the Minor program 'Child' of the faculty of Health of Amsterdam University of Applied Sciences) and Deborah Cabal, Marieke Hagemeijer, Karin Stellingwerf (former students of bachelor 'Exercise therapy' of Amsterdam University of Applied Sciences) and Gerben ter Riet for his advice on the statistical analysis of the data.

## Author Contributions

**Conceptualization:** Helga Haberfehlner, Liesbeth de Vries, Margo J. van Hartingsveldt.

**Data curation:** Helga Haberfehlner.

**Formal analysis:** Helga Haberfehlner.

**Investigation:** Helga Haberfehlner, Liesbeth de Vries.

**Methodology:** Helga Haberfehlner, Liesbeth de Vries, Edith H. C. Cup, Imelda J. M. de Groot, Maria W. G. Nijhuis-van der Sanden, Margo J. van Hartingsveldt.

**Project administration:** Helga Haberfehlner, Liesbeth de Vries, Margo J. van Hartingsveldt.

**Writing – original draft:** Helga Haberfehlner, Liesbeth de Vries, Margo J. van Hartingsveldt.

**Writing – review & editing:** Helga Haberfehlner, Liesbeth de Vries, Edith H. C. Cup, Imelda J. M. de Groot, Maria W. G. Nijhuis-van der Sanden, Margo J. van Hartingsveldt.

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
