## [Decision Letter · Decision Letter 0]

1 Dec 2022

PONE-D-22-29322Reference data for handwriting readiness assessment – using Writing Readiness Inventory In Context (WRITIC) and fine-motor coordination tests

PLOS ONE

Dear Dr. de Vries,

Thank you for submitting your manuscript to PLOS ONE. After careful consideration, we feel that it has merit but does not fully meet PLOS ONE’s publication criteria as it currently stands. Therefore, we invite you to submit a revised version of the manuscript that addresses the points raised during the review process.

Thank you for submitting your valuable work.

The reviews, which are insightful and interesting, pointed to some relevant aspects. The authors will notice the reviewers found merits in your study, but also raised several important concerns.

By my own reading, the manuscript needs a little bit of refinement, mostly related to conciseness and stats.

1) I think the Title could be a little bit more intuitive. I have no concern regarding the way it is, but you can a more "punchy" title to gather other's attention to it;

2) In abstract, please check some important issues: (i) provide mean age and SD for your sample in Methods instead of Results; (ii) do not begin the sentence with numbers, the authors need to change "Results: 374" to "Results: Three hundred and seventy four" and this needs to be constant thorough the text; and (iii) the WRITIC scores range (from x to x, for example);

3) Your eligibility criteria need to be better emphasised;

4) Based on the skewed sample, this reminds me of the presentation of kurtosis and (why not?) conduct log-transform or fractional rank?

5) Please clarity why you only provided descriptive of WRITIC and nothing on exploratory factor analysis? Was WRITIC previously validated in the same sample? If yes, what is the novelty of the study? If no, why can't the authors provide EFA as a sup. file for this specific sample?

6) Please avoid the use of paragraphs without proper references in Discussion. And check outdated references to ensure transparency and an updated manuscript.

Finally, please double check English and the refs. list accordingly to the Journal's standard.

Please respond AND highlight all comments.

We look forward to receiving your revised manuscript.

Kind regards,

Thiago P. Fernandes

PLOS ONE

Journal Requirements:

**Additional Editor Comments:**

Please read my comments.

Reviewers' comments:

Reviewer's Responses to Questions

**Comments to the Author**

1. Is the manuscript technically sound, and do the data support the conclusions?

Reviewer #1: Partly

Reviewer #2: Yes

2. Has the statistical analysis been performed appropriately and rigorously? 

Reviewer #1: Yes

Reviewer #2: Yes

3. Have the authors made all data underlying the findings in their manuscript fully available?

Reviewer #1: No

Reviewer #2: Yes

4. Is the manuscript presented in an intelligible fashion and written in standard English?

Reviewer #1: Yes

Reviewer #2: Yes

5. Review Comments to the Author

Reviewer #1: Reference data for handwriting readiness assessment – using Writing Readiness

Inventory In Context (WRITIC) and fine-motor coordination tests

PONE-D-22-29322

Reviewer Becher

The manuscripts describes the results of the WRITEC in a typical developing population children, mean age 5,6 yrs (SD 0,4 yrs) of a Dutch population. To get data about the validity, the nine-hole peg test and the Timed Test in Hand Manipulation were scored.

Percentile scores were calculated, children with a score below the 15th percentile were supposed to be at risk for developing writing problems at the start of learning writing.

Abstract

The abstract is clearly written.

P8 S 40

The concusion “Conclusion: The reference data of the WRITIC allow assessing which children are at risk to develop handwriting difficulties.” Does not fully fit the contents: low performance is described, but the relation with writing problems later in the development has not been demonstrated: the are possibly at risk.

P 9 s58

“in previous studies …. Etc” In these studies, only statistical analysis has been performed for groups. `the results from group studies is translated to individual risk. That is not valid: other statistics are used to predict the chance for an individual to get problems with learning writing, with a confidence interval. So, I propose to add “ group studies”.

P10 S 67-75

The tests described are all “laboratory” tests (capacity tests) of a specific construct, supposed to be important to learn writing. However, for all these tests it is not demonstrated on individual level to what extend they really predict problems with learning writing, with specificity and sensitivity. I proposed to add a remark that these capacities are supposed to have a relation with the development of writing problems, but the predictive value is unknown.

The participants and methods section is clearly described.

P 12 S 118: Cronbach’s alpha?

Results are properly described.

About the characteristics of the study population, nothing is mentioned about the background of the children. In the Netherlands, a lot of non-native families are present, in big cities up to 60% speak at home other languages than Dutch. There are cultural differences in raising up children, participation in kindergarten and playing at home.

Are the reference values mainly based on results of children of white Dutch families?

Discussion

P19 S 260

“No difference was found for age groups for the 261 performance on the WRITIC-TP,”

The authors do not comment on this remarkable findurthering: if the WRITIC-TP has a predictive value for learning writing, you expect higher scores with age. How can the authors explain this finding?

Conclusion

P22 S 326

Children are possibly at risk for

P22 S327-9

“When children have minor difficulties according to the WRITIC-TP, kindergarten teachers can be advised and supported in training the children in the mastery of prewriting skills to prevent handwriting difficulties in higher grades.”

The study does not provide any evidence for this statement and should be skipped.

Further research is needed to investigate the predictive value of a low WRITIC-TP score for development of writing problems at school.

The remark that occupational therapy is indicated is also beyond the contents of the research, please stick to the results of your results and remove personal statements.

Reviewer #2: The work presents important results for interventions and the creation of strategies that assist in child development, mainly ensuring the standardization of measures that can be used mainly as screening in the school context. I would like to congratulate the authors because this type of work is extremely important and aids in child development through a multidisciplinary dialogue.

In general, the work presents current references (following the APA standard), objective language, and robustness of data analysis. The following highlights some suggestions for possible changes aimed at the authors.

Introduction

This section is very objective and presents relevant topics for understanding the importance of writing learning processes.

However, I recommend adding on the benefits that arise from these processes and deepening how difficulties in this sense can influence development globally.

I also recommend deepening the "negative side effects in later series" cited on page 3 of the manuscript.

In the introduction, it is important that the reader can glimpse possible connections between manual writing and integration with other aspects evaluated, such as visuomotor skills. Thus, I recommend the addition of a paragraph that presents some studies that have evidenced these relationships and the importance of developing these pre-handwriting skills.

Materials and methods

In the text, the ethical aspects of research and data collection are adequately presented.

However, I recommend deepening in relation to the exclusion criteria used.

In the abstract, it is stated that no data had previously been collected using the instruments chosen with a Dutch sample.

However, it is unclear whether such instruments had previously been adapted for this population. In order to avoid possible misunderstandings regarding the application and use of the instruments, I recommend adding a brief sentence quoting the versions used in the work and clarifying these previously highlighted points.

Procedure

In the text, it is highlighted that the "Timed-TIHM and 9-HPT were administered in the same session outside the classroom in an individual situation." Since the instruments were applied in the same session, I recommend adding more details about how the session was handled—could the child request to stop during the application? Was fatigue considered a variable that could influence?

Statistical analysis

The authors presented in an appropriate and explanatory way the analyzes used, in this section, I only recommend the addition of a reference highlighted in the manuscript.

Results

The findings were presented adequately, both in text format and in the table.

Discussion

The importance of considering cultural and teaching differences from other countries in future applications and study replications is addressed. I believe it is critical to emphasize the characteristics of Dutch teaching that the authors believe differ from other studies and how this influences the results obtained.

I recommend including a higher resolution version of image 1 in particular.Regarding image 2, I recommend changing the flowchart format or transposing the data to a table.

In general, the work presents precisely and adequately the results obtained in the application of the three instruments used. The findings are relevant for future interventions, especially in the school environment. As a result, I recommend that the manuscript be approved after minor revisions.

6. PLOS authors have the option to publish the peer review history of their article (what does this mean?). If published, this will include your full peer review and any attached files.

Reviewer #1: **Yes: **Em. prof. Jules Becher

Reviewer #2: No

---

## [Author Response · Author response to Decision Letter 0]

26 Jan 2023

Thank you very much for reviewing our manuscript and the positive and very constructive feedback of the reviewers. We have added a response to reviewer letter to answer in detail and explain the changes made. We hope you enjoy reading our revised article.

---

## [Decision Letter · Decision Letter 1]

9 Feb 2023

PONE-D-22-29322R1Ready for handwriting? A reference data study on handwriting readiness assessmentsPLOS ONE

Dear Dr. de Vries,

I really appreciated your valuable and thoughtful edits. The remaining concerns were addressed, but one reviewer requested a very quick-to-solve thing to be addressed. I call out the authors’ attention to address this concern as soon as they can, so we can proceed fast with your study. Also I need to state that the manuscript reads better now and still concise & straightforward. I wish success with the study. 

The reviewer’s comment:

Ref 3: in summary: “Preliterate, five-year old children printed, typed, or traced letters and shapes, then were shown images of these stimuli while undergoing functional MRI scanning. A previously documented "reading circuit" was recruited during letter perception only after handwriting-not after typing or tracing experience. These findings demonstrate that handwriting is important for the early recruitment in letter processing of brain regions known to underlie successful reading. Handwriting therefore may facilitate reading acquisition in young children.”

The reading circuit was only activated by handwriting in healthy children.

However, there are dyslexic children who can read at normal level but are of very low level in writing / spelling. Children with weakness of the arm /hand muscles are able to learn writing at normal level for their age.

S50  “Handwriting is essential for 51 learning reading and spelling.”  This statement is not true, and also not supported by the reference: yes, it may facilitate reading, but is not a condition to be able to read. Please skip this sentence: S 54 is enough as statement: “ Handwriting may facilitate learning reading and spelling.”

P37 S98-99: “WRITIC-TP, Timed-TIHM 99 and 9-HPT”  Why did you change the order of the tests in comparison to the introduction before?  You keep this order in the “Instruments” section, so please change the order in the introduction.

Thank you for submitting your manuscript to PLOS ONE. After careful consideration, we feel that it has merit but does not fully meet PLOS ONE’s publication criteria as it currently stands. Therefore, we invite you to submit a revised version of the manuscript that addresses the points raised during the review process.

We look forward to receiving your revised manuscript.

Kind regards,

Thiago Fernandes, PhD

Academic Editor

PLOS ONE

Journal Requirements:

Reviewers' comments:

Reviewer's Responses to Questions

**Comments to the Author**

1. If the authors have adequately addressed your comments raised in a previous round of review and you feel that this manuscript is now acceptable for publication, you may indicate that here to bypass the “Comments to the Author” section, enter your conflict of interest statement in the “Confidential to Editor” section, and submit your "Accept" recommendation.

Reviewer #1: All comments have been addressed

Reviewer #2: All comments have been addressed

2. Is the manuscript technically sound, and do the data support the conclusions?

Reviewer #1: Yes

Reviewer #2: Yes

3. Has the statistical analysis been performed appropriately and rigorously? 

Reviewer #1: Yes

Reviewer #2: Yes

4. Have the authors made all data underlying the findings in their manuscript fully available?

Reviewer #1: Yes

Reviewer #2: Yes

5. Is the manuscript presented in an intelligible fashion and written in standard English?

Reviewer #1: Yes

Reviewer #2: Yes

6. Review Comments to the Author

Reviewer #1: (No Response)

Reviewer #2: First, I would like to thank the authors for the changes made to the manuscript according to the previously indicated suggestions. The addition of new information about the writing learning process enriched the discussion and justified the study. Similarly, the addition of the exclusion criteria and more details regarding the adapted instrument and application procedure indicated a process of transparency in data collection.

In relation to other suggestions previously indicated, I emphasize that the new title is adequate and the changes in the abstract are substantial. Considering the current version of the manuscript, I suggest accepting it for publication in the journal.

7. PLOS authors have the option to publish the peer review history of their article (what does this mean?). If published, this will include your full peer review and any attached files.

Reviewer #1: **Yes: **Em. Prof. Jules Becher

Reviewer #2: No

---

## [Author Response · Author response to Decision Letter 1]

13 Feb 2023

Thank you very much for your quick review and positive evaluation. Thank you so much for giving us the opportunity to present to you our manuscript with minor revisions.

---

## [Editor Report · Decision Letter 2]

16 Feb 2023

Ready for handwriting? A reference data study on handwriting readiness assessments

PONE-D-22-29322R2

Dear Dr. de Vries,

We’re pleased to inform you that your manuscript has been judged scientifically suitable for publication and will be formally accepted for publication once it meets all outstanding technical requirements.

Kind regards,

Thiago P. Fernandes, PhD

Academic Editor

PLOS ONE

Additional Editor Comments (optional):

Thank you for your submission. Wishing you success with the study
---

## [Editor Report · Acceptance letter]

23 Feb 2023

PONE-D-22-29322R2 

Ready for handwriting? A reference data study on handwriting readiness assessments 

Dear Dr. de Vries:

I'm pleased to inform you that your manuscript has been deemed suitable for publication in PLOS ONE. Congratulations! Your manuscript is now with our production department. 

Kind regards, 

on behalf of

Dr. Thiago P. Fernandes 

Academic Editor

PLOS ONE